Enhanced payload volume in the least significant bits image steganography using hash function

http://orcid.org/0000-0002-7121-495X Ghadi Yazeed Yasin 1
AlShloul Tamara 2
Nezami Zahid Iqbal 3
Ali Hamid 4 hamid.ali@ntu.edu.pk
http://orcid.org/0000-0003-1839-2527 Asif Muhammad 4
Jaward Bah Mohamed 5
1 Department of Computer Science/Software Engineering, Al Ain University , Al Ain , UAE
2 College of General Education, Liwa College , Abu Dhabi , UAE
3 Department of Computer Science, The Superior University , Lahore , Pakistan
4 Department of Computer Science, National Textile University , Faisalabad , Pakistan
5 Zhejiang Lab , Zhejiang , China
Cunkas Mehmet
Electronic publication date: 2023 Nov 13
Publication date: 2023
Volume: 9
Electronic Location ID: e1606
Received 2023 Jul 12; Accepted 2023 Aug 29
Copyright: © 2023 Ghadi et al.
Copyright year: 2023
Copyright holder: Ghadi et al.
License: This is an open access article distributed under the terms of the Creative Commons Attribution License, which permits unrestricted use, distribution, reproduction and adaptation in any medium and for any purpose provided that it is properly attributed. For attribution, the original author(s), title, publication source (PeerJ Computer Science) and either DOI or URL of the article must be cited.
License URL: https://creativecommons.org/licenses/by/4.0/

Keywords: Steganography, Hash function, High payload, Caesar cipher

Funding: The authors received no funding for this work.

==============================
The art of message masking is called steganography. Steganography keeps communication from being seen by any other person. In the domain of information concealment within images, numerous steganographic techniques exist. Digital photos stand out as prime candidates due to their widespread availability. This study seeks to develop a secure, high-capacity communication system that ensures private interaction while safeguarding information from the broader context. This study used the four least significant bits for steganography to hide the message in a secure way using a hash function. Before steganography, the message is encrypted using one of the encryption techniques: Caesar cipher or Vigenère cipher. By altering only the least significant bits (LSBs), the changes between the original and stego images remain invisible to the human eye. The proposed method excels in secret data capacity, featuring a high peak signal-to-noise ratio (PSNR) and low mean square error (MSE). This approach offers significant payload capacity and dual-layer security (encryption and steganography).

Introduction

Today, the internet plays a vital role. Internet security and online communication are essential topics that catch people’s attention. Since humans are social by nature, we talk to each other in various situations. Everyone has their favorite way of communicating, and sometimes we share private information with the people we mean to share it with Delenda & Noui (2018). The demand for ensuring the authenticity of data and finding efficient methods to protect its integrity has been on the rise. This requirement has gained greater significance owing to the inherent simplicity of altering digital data. In this context, steganography can replace encryption and watermarking for safer data transmission and to maintain privacy (Shyla, Kumar & Das, 2021). Data transmitted between a sender and recipient gains enhanced security through steganography and cryptography. Steganography involves concealing sensitive information within public files to avoid visual detection. Previous studies (Islam et al., 2021; Khan et al., 2016) reveal steganography’s capacity to obscure unusual information within confidential or secret content. It is important to note that steganography and cryptography can sometimes be confused as they influence how personal information is managed. Their crucial distinction lies in steganography’s use of sensitive data, while the other technique seemingly lacks concealment. This distinction resonates with the notion that individuals and groups often avoid engaging with information they believe is confined within them (Voleti et al., 2021). The term “steganography,” with its roots in Greek, originates in “Steganos.” This name is formed by combining “secret” and “graphic,” which symbolize the concept of “writing.”

On the other hand, steganography hides data like an image, text, video, or piece of music. It’s commonplace to use the words “steganography” and “cryptography” interchangeably. Watermarking verifies the message’s integrity, steganography hides it, and cryptography muddles it. Before concealing in steganography, select a suitable carrier. Choosing a trustworthy steganographic technique that can encrypt genuine information is necessary. The secret message can then be delivered to the recipient via any modern communication method by the sender. When the message is received, the recipient should use the proper extraction process to uncover the hidden information. A range of appropriate steganographic techniques is used to ensure security, depending on the kind of carrier (Sasmal & Mula, 2021). These are as follows. Video steganography: As the security requirements for transmitting secret messages become more stringent and video becomes more popular. Video steganography is increasingly important as a study field in various data-concealing technologies.

Image steganography: Employing an image as the cover object for steganography. The majority of these techniques use the image’s pixel intensities to mask the information.

Network protocol steganography: A network protocol, like TCP, UDP, ICMP, or IP, is used as a cover object in network protocol steganography. In this situation, steganography can be achieved using unused network protocol header bits.

Text steganography: employs various methods, such as upper-case letters, white spaces, and the number of tabs, which is akin to Morse code, to disguise information.

Audio steganography: Audio as a carrier for information concealment is known as audio steganography. Due to its ubiquity, voice-over-Internet protocol (VOIP) has become a significant medium. Digital audio formats used in this steganography include WAVE, AVI, MIDI, MPEG, and others.

The following elements make up the fundamental building blocks of steganography. 1) Cover image: The cover image conceals secret information. It hides images or data.

2) Secret image: The “secret image” is the picture or information that needs to be hidden in the “cover image.” It can be any picture, word, or other digital data meant to stay secret.

3) Stego image: Steganographically integrating the secret image or information into the cover image creates the stego image. The stego image resembles the cover but conceals information.

4) Embedding and extracting algorithms: The embedding algorithm embeds the hidden picture or data into the cover image without being noticeable. The complementary extraction method recovers the secret picture or data from the stego image.

5) Stego key (optional): Some steganographic systems employ stego keys. The embedding and extracting methods use the stego key. It may secure, authenticate, or restrict access to concealed information.

Image steganographic algorithms are assessed using three key criteria (Ahmad et al., 2022).

Security: Steganographic algorithms must be secure. It tests the algorithm’s ability to hide secret information and withstand assaults and discovery. The algorithm should protect embedded data from unauthorized access. Statistical analysis, ocular examination, and steganalysis are widely used to assess algorithm security.

Capacity: The amount of secret data concealed in the cover image is called capacity. It tests the algorithm’s ability to hide much information in the cover image without degrading the visual quality. The algorithm’s data embedding rate, secret payload size, and image quality are evaluated to determine capacity.

Robustness: Robustness is a steganographic technique’s ability to survive image processing, transmission problems, and assaults without losing data. A robust steganographic approach should preserve and retrieve secret information even after standard procedures or distortions.

Since the limited payload capacity sends a low-quality image to the receiver side, this results in a loss of information at the receiver’s end, leading to unclear embedded images. Therefore, it is imperative to incorporate high payload data on the sender’s side. Consequently, transmitting high-quality images becomes achievable on the receiver’s side (Zheng et al., 2016).

Therefore, in this article, we proposed a data-embedding approach: To enhance the payload and

To secure confidential information in a specified way in the cover image with the slightest modification.

The cover image’s quality is undetectable to the human eye.

The rest of the article is structured as follows. “Related Work” describes related work. The suggested strategy is given in “Proposed Approach”. “Experimental Description And Results” presents the experiments and findings. The article is concluded in “Conclusion”.

Related work

Protecting systems and data is vital due to complex software packages and evolving technologies that threaten data security, integrity, and availability. In Ali, Sohrawordi & Uddin (2019), the author proposed a unique steganography technique inside RGB shading space to enhance security over existing solutions. The authors employ various picture quality assessment procedures, such as MSE, MAE, PSNR, SSIM, and NCC (normalized correlation coefficient). The results demonstrate strength, invisibility, and security compared to earlier techniques. The suggested method obtained an average score of 03.67% better for PSNR correlation. In Rahman et al. (2020), the authors present a steganography categorization scheme for professional and non-professional domains. Steganography involves essential factors like SSIM, NCC, image fidelity (IF), and resilience. Examining Stego images, payload, MSE, and correlation brings to light interrelated challenges and their significance. This study aims to investigate and contrast various steganography algorithms utilizing traits like PSNR, MSE, and robustness. In Dhawan & Gupta (2021), the authors divided steganography into technical and non-technical categories and categories based on its use. The study aims to evaluate and contrast various steganography algorithms utilizing parameters including resilience, PSNR, and MSE. Based on these features and analysis of all issues, this study recommends enhancing a steganography system, aiming for stego images of exceptional quality, increased payload capacity, and improved reliability. In Charan et al. (2015), the author suggests an innovative method that involves converting ordinary text to encrypted text and then inserting it into a colour picture. There are two phases to the encryption process: the 1st phase uses the Caesar cipher method, and the 2nd uses chaos theory. The 3, 3, 2 LSB replacement procedure is used to incorporate the cipher text retrieved after encryption.

In Wade et al. (2019), the authors compared the stego and cover images using the following metrics: peak signal-to-noise ratio, CPU time, mean square error, histograms, structure similarity index measures, and feature similarity index measurements. Their research and experiments reveal that their proposed approach is faster and more efficient than traditional LSB methods. In Muhammad et al. (2015), the authors show how grayscale modification (GLM) is used to hide secret information in the cover image and how multilevel encryption (MLE) encodes the secret data. In the GLM technique, the current pixel stays intact if the sensitive bit is zero and the cover pixels exist even, or vice versa. If there is an odd number of cover pixels and the secret bit is 0, then the current cover pixels decrease by 1. If the number of cover pixels is even and the secret bit is greater than 1, then the current cover pixel quantity is increased by 1. The embedded payload is relatively small, yet this strategy improves the PSNR of the stego image.

In Shehzad & Daǧ (2019), the authors proposed a method in which the cover image is split into four-pixel chunks that do not overlap (2 × 2). Each block’s decimal numbers are used to compute the determinant value for that block. A hidden data bit is inserted into a block of pixels using the determinant method. The determinant method used only one block to hide one secret data bit. This technique enhances the stego image quality but has a limited embedding capacity since only one bit may be hidden per block. In Muhammad et al. (2017), the authors concealed secret information in the cover image using stego key-directed adaptive least significant bit (SKA-LSB) replacement and the Multilevel Encryption Algorithm (MLEA) to encrypt confidential information. SKA-LSB performs an XOR operation on the cover pixel’s red channel key and LSB. If the sum of the remaining bits is 1, include the secret information bit in the LSB of the green channel. If not, have the confidential information to the LSB of the blue channel. Encapsulation of secret information is only one bit per pixel, but this strategy increases his PSNR value for stego images.

In Kuo, Kuo & Wuu (2015), the Run-Length Encoding (RLE) methodology and the Multi-Bit Generalised Exploding Modification Direction (MGEMD) algorithm were suggested by the authors as a way to conceal information concealed in the cover picture. In Liu, Chang & Chien (2017), the authors proposed a revolutionary approach that relies on LSB substitution and places two bits of hidden data into one cover pixel following the mapping strategy. Three bits are modified in the cover pixel. By using this method, the stego image quality is improved by PSNR 49.04 dB. In Hong (2018), the authors utilize the modified quantization level approach to its fullest extent to enhance stego image performance and effectiveness. This technique divided blocks into easy and complicated blocks using a particular threshold. Smooth blocks hold more sensitive data, but complex blocks only store one bit per block. The Adaptive Pixel Pair Matching (APPM) technique reduces distortion by modifying the block. This method’s flaw is its low embedding capacity, 32.27 dB PSNR.

In Setiadi & Jumanto (2018), the authors proposed a hybrid approach combining the Canny and Sobel edge detection algorithms. Edge detection methods Sobel (S) and Canny (C) are utilized to locate the edges of the cover image. The result is saved as CSea by combining the outputs of both algorithms. The LSB approach inserts hidden bits in the cover image’s structuring element. If the edge area does not have enough space to display the hidden message, the secret information will be kept in a smooth place. Although the embedding capability of secret data in cover images is quite limited, this strategy does increase the quality of stego images. In Islam et al. (2016), the authors proposed a method in which the colour area of the cover pixels is divided into high and low levels. Cover pixels contain more hidden bits of the low-level colour spectrum than high-level colour pixels. When the pixel ranges lie between 0–31, 32–63, 63–127, and 28–155, respectively, Cover pixels contain secret data of size 4, 3, 2, and 1 bit. This approach enhances the PSNR value of the stego image, but the embedded payload of the hidden info is minimal. In Shehzad & Dag (2017), the authors proposed a method that splits cover pixels’ 3–7 MSBs into four pairs of two bits and secret information bits into pairs of two bits. The matching pair number adjusts each cover image’s two LSBs after comparing one secret pair with each of the four covering pixels. This method raises the stego image’s PSNR value. In Khalind & Aziz (2013), the authors introduced a technique that divides a cover image into edge and smooth pixels, each holding varying levels of information. This method calculates an adaptive parameter K for each pixel to include hidden details, influencing the stego image’s quality. The value of K affects PSNR, resulting in high PSNR for low K values when embedding secret information in the cover image and vice versa. In Mohammed & Mohamed (2016), the authors used a straightforward LSB substitution method; secret information is inserted into cover pixels dependent on the value of a variable K ranging from 2 to 5. This method replaces the K LSBs of the cover pixel’s K secret data bits. The value of the variable K determines the stego image’s quality. PSNR will be small if secret information bits are covered up with a large value of K, and vice versa. Based on PVD and LSB, in Lu & Lu (2017), the authors proposed a method to divide a grayscale image into 1, 2, and 3 predetermined blocks of every three pixels to improve the embedding capacity. The FPPD approach (five-pixel pair difference) with a fixed block size of 2 × 3 is enhanced by this method. Using the FPPD technique, the stego image quality is adequate, but the secret information payload is improved to three bits per pixel. In Swain (2016), the authors separate the covering pixels into blocks of four pixels that do not overlap (2 × 2).

The hybrid PVD/LSB Substitution technique incorporates the secret data in the cover picture. Although the confidential information encapsulates less than one bit per pixel, this strategy increases the stego image’s PSNR value. In Liao, Wen & Zhang (2011), the authors split the cover image into 4 × 4 non-overlapping sections. The four sub-blocks that comprise each block are top left, bottom left, top right, and bottom right. Every subblock has four pixels (2 × 2). As a result, the edge and sleek pixels are separated from each block’s pixels. The edge area pixels contain more hidden bits than the non-area pixels. The embedding payload and tolerable PSNR are both improved.

In Hussain et al. (2018), the authors split the cover pixels into blocks of high-level and low-level pixels. Each of the four Subblocks Each method (pixel value difference, PVD shift, prediction error correction, and MPE) used to embed the encrypted information into the high-level block has its strengths and weaknesses. The LSB alteration embeds the scrambled information in the sub-blocks that make up each block (top left, bottom left, top right, bottom right). The secret data payload’s resistance to attacks is increased with this technique while keeping adequate stego image quality. In Tseng & Leng (2014) author scales the cover image using an interpolation algorithm and employs LSB replacement to cover up secret information. The user can zoom in or out on his cover photo to better embed sensitive information in the cover photo. This technique enhances the embedding payload while maintaining decent PSNR.

In Khodaei, Sadeghi Bigham & Faez (2016), the authors split the covering pixels into two-bit blocks that follow one another without overlapping. The secret data is inserted into the cover image via LSB replacement if both pixels in the block have the lowest value. Otherwise, his PVD of two pixels is calculated. After that, the secret information is placed in the cover pixels using LSB substitution and the PVD value. This technique enhances stego image quality while also improving the embedding payload. In Bai et al. (2017), using an edge detection technique, the authors divide the cover pixels into “edge and smooth pixels areas”. The LSB replacement approach then includes concealed information in the cover picture. Compared to a smooth region, the edge area’s pixels are employed to carry additional hidden bits. This technique enhances the hidden data payload while maintaining high-quality stego images. In Zakaria et al. (2018), the author splits the cover pixel’s MSBs into two pairs of two bits and secret message bits into pairs of two bits. Four possible matches between secret and pixel pairs are created by comparing two pairs of cover pixels with two pairs of secret data. The cover image’s two lower-case LSBs are modified according to the matched pair number. With strong PSNR, this technique increases the ability to insert hidden data in cover images. In Hamza et al. (2021), the authors developed a hybrid steganography approach that uses LZW compression to compress data in the first phase, and then the AES cryptographic algorithm is used. Image steganography based on pattern matching is developed using LSB substitution. Four pairs of cover pixels and four pairs of two-bit secret information are compared to generate 16 potential matching instances. Each pixel’s four LSBs are adjusted according to how closely the two numbers match. This method can carry eight hidden data bits within a pixel, eight or 10 secret data bits between two pixels, and, at worst, four hidden data bits between two pixels.

Proposed approach

The suggested strategy uses two things—message encryption techniques and embedding algorithms.

Massage encryption technique

A variety of cryptographic algorithms are utilized to secure the data. Some standard encryption techniques include:

Symmetric-key cryptography: When a secret key is used for encryption and decryption operations, the technique is called symmetric key cryptography, also called symmetric encryption. Symmetric encryption is the most common type in email and online banking applications. The advantage of using a symmetric key is that it is easy to encrypt and decrypt data using the same key. Additionally, symmetric-key cryptography is relatively simple to implement, making it a good choice for applications without critical security. However, symmetric-key cryptography has one major drawback: it can be easily hacked if the key is compromised.

CaEsAr cipher

A kind of symmetric-key cryptography known as the Caesar cipher employs a substitution cipher. Established in the 1st century BC. Created by Julius Caesar and is still in use today. To make a text message difficult to interpret, the Caesar cipher, which bears the name of its creator, operates by replacing arbitrary words with the message’s characters. Caesar cipher Considered one of the safest ways available, this encryption safeguards critical information like passwords and bank data. This method shifts each character in the message by the number of alphabetic positions. Shift three, and the letter b becomes an e. 1) Get the secret massage

2) Use the encoding algorithm (Caesar cipher)

3) Keep your cipher massage

Vigenère cipher

A polyalphabetic substitution cipher known as the Vigenère cipher replaces every alphabetic letter with a distinct letter using a table of key values, as illustrated in Fig. 1. Francesco Vigenère devised an algorithm for it in 1,694, and his name was given to the cipher. Polyalphabetic ciphers are less vulnerable to cryptanalysis. The polyalphabetic cipher is ideal for encrypting material that parties outside the sender and recipient will read. The Vigenère cipher, constructed on a straightforward replacement algorithm, substitutes one of the 26 continuous letters of the alphabet for every letter of the alphabet. The initial stage of encryption of a message using this cipher is to select key values (26 total) that specify the characters to replace corresponding characters in the message. The table below shows key values and associated characters for both English and Latin scripts.

Figure 1 Vigenère table values.

Asymmetric-key cryptography: To prevent unauthorized access or use, asymmetric cryptography, often known as public-key cryptography, encrypts and decrypts data using a pair of related keys (a private key and a public key).

English letter values in key the alphabet is A B C D E F G H I J K L M N O P Q R S T U V W X Y Z. Latin letters and key values 1 2 3 4 5 6 7 8 9 10 11 12 13 14 15.

Hash function

Hash functions are mathematical procedures for identifying any object with a unique identifier. A hash function produces a consistent-length output derived from an input. Each input processed by a hash function yields a unique output. The identical hashing technique for generating the identifier can also validate data integrity. This attribute proves advantageous in various applications, including file identification, database access, and password hashing.

This article employs a security approach centered around a hash function. The method utilizes a key and a pixel number as inputs and generates an output number. Before communication, the sender and receiver have already exchanged the designated key value. We illustrate to clarify this:

To conceal four message bits within pixel 50 of the cover picture, let’s assume a key value of 11. Upon applying the hash function, this key yields the output number 6. Subsequently, using the hash function with a value of four signifies the desire to hide the message within the final four bits originating from this generated bit, resulting in “2.” Incrementing this value by “1,” the sender replaces the message bits with circularly positioned pixel bits 6, 5, 8, and 7 of pixel 50. Similarly, for values 0, 1, and 3, the sender follows an analogous procedure, utilizing the respective pixel bits for replacement:

Case 0: 8, 7, 6, and 5;

Case 1: 7, 6, 5, and 8;

Case 3: 5, 8, 7, and 6;

The recipient extracts the message bits similarly.

Embedded algorithm

The user on the sender side reads the cover image and the text message generated in “Massage Encryption Technique” as input, producing a stego image as output. The procedure starts by reading the cover image and the encrypted text messages. To secure the message, first, apply the Vigenère cipher or the Caesar cipher encryption methods. Read the whole cover image pixel by pixel and hide the encrypted text using the hash function. The text is hidden circularly in each pixel of the cover image. So, it is hard to access by the third person. Figure 2 illustrate how the embedded algorithm functions on the sender side. The detail is given in Algorithm 1.

Figure 2 Encoding flow diagram.

Table 4 Gray stego image of forest.

Algorithms	PSNR	MSE	Payload	Bits per pixels (bpp)	
Dhawan & Gupta (2021)	51.99	0.41	21,845	0.08	
Wade et al. (2019)	34.76	21.73	87,381	0.33	
Muhammad et al. (2015)	44.57	2.27	222,054	0.8	
Shehzad & Daǧ (2019)	52.12	0.39	262,144	1	
Muhammad et al. (2017)	33.42	29.58	262,144	1	
Kuo, Kuo & Wuu (2015)	48.24	0.97	285,736	1.09	
Setiadi & Jumanto (2018)	37.5	11.56	589,824	2.25	
Islam et al. (2016)	44.4	2.36	589,824	2.25	
Shehzad & Dag (2017)	38.08	10.89	778,567	2.97	
Hussain et al. (2018)	37.25	12.24	810,024	3.05	
Mohammed & Mohamed (2016)	39.79	10.81	805,492	3.07	
Lu & Lu (2017)	38.56	9.05	810,501	3.09	
Khalind & Aziz (2013)	38.33	9.61	815,912	3.11	
Liao, Wen & Zhang (2011)	37.13	12.59	822,042	3.13	
Swain (2016)	33.61	28.31	828,375	3.16	
Khodaei, Sadeghi Bigham & Faez (2016)	33.44	29.44	854,796	3.26	
Proposed	37.97	10.38	262,144	4	
34.93	20.90	524,288	4	
33.12	31.69	786,432	4	
31.85	42.40	1,048,576	4	

Sender side

On the sender’s end, Algorithm 1 is employed. Hide an encrypted ciphertext message inside the cover image.

Extracted algorithm

At the receiver’s end, the concealed information is deciphered. The user takes the stego image generated on the sender side and the corresponding key to extract the text message. This process begins by reading the stego image and utilizing the key to access the embedded text message. The concealed message is unveiled by employing the appropriate decryption technique, such as the Vigenère cipher or the Caesar cipher. The stego image is analyzed pixel by pixel to recover the encrypted text, which was hidden using the hash function during the sender’s procedure. Figure 3 illustrate how the embedded algorithm functions on the sender side. The detail is given in Algorithm 2.

Figure 3 Decoding flow diagram.

Algorithm 1 High payload algorithm based on hash function on the sender side.

    Input: The algorithm will accept the Cover Image and the Text Message generated in III-A as input.	
    Outcome: This algorithm produces a securely encrypted Stego Image as a result.	
   Initiate the process	
1    Read cover image and encrypted text messages.	
2    Protect the message by utilizing the encryption method Vigenère cipher or Caesar Cipher.	
3    Navigate within the cover image.	
4    Iterating over the height of the cover image (i = 1 to height)	
5      For each width of the cover image (j = 1 to width)	
6       If there are still unrevealed text bits:	
7        Use a hash function to find the current pixel’s Least Significant Bit.	
8         Calculate SB1 using the modulo operation, incorporating the current height (i − 1) and width (j) of the image, ensuring the result is confined within the range of 0 to 10	
9         Determine SB by applying the modulo operation to SB1, ensuring the result lies within the range of 0 to 3	
          If SB is equal to 0	
           Replace pixel bits 8, 7, 6, and 5 sequentially with four corresponding message bits.	
          If SB is equal to 1	
           Replace pixel bits 7, 6, 5, and 8 sequentially with four corresponding message bits.	
          If SB is equal to 2	
           Replace pixel bits 6, 5, 8, and 7 sequentially with four corresponding message bits.	
          If SB is equal to 3	
           Replace pixel bits 5, 8, 7, and 6 sequentially with four corresponding message bits.	
10        Conclude by incorporating the freshly computed value into the ongoing state of the output pixel.	
11       End of the conditional statement	
12      End of the nested loop for width	
13    End of the loop for height	
14    Calculate the Mean Squared Error (MSE) by employing pixel data from the cover and stego images.	
15    Compute the Peak Signal-to-Noise Ratio (PSNR) using the calculated MSE.	
16    Preserve the Stego image.	
17  End	

Algorithm 2 High payload algorithm, based on hash function on the receiver side.

    Input: The algorithm takes a Stego Image as its input.	
    Output: It generates a Text Message embedded within the Stego Image.	
   Initiate the process	
1    Upon establishing the size of the text message as S, compute S1 as the size of the bit stream (S * 8)	
2    Create an empty bit-string denoted as M, referred to as “extracted bits.”	
3    Traverse through the provided Stego image.	
4    Iterating over the height of the stego image (i = 1 to height)	
5     Iterating over the width of the stego image (j = 1 to width)	
6      If (there are still unretrieved message bits.)	
7       Utilize the sender-side hash function to ascertain hidden message bits from the current pixel.	
8       Calculate SB1 as mod ((height × (i−1) + j), 11)	
9       Determine SB by calculating mod (SB1, 4)	
         If SB equals 0	
          Sequentially store the 8th, 7th, 6th, and 5th-bit values from the pixel under consideration into the bit string M (extracted bits)	
         If SB equals 1	
          Sequentially store the 7th, 6th, 5th, and 8th-bit values from the pixel under consideration into the bit string M.	
         If SB equals 2	
          Sequentially store the 6th, 5th, 8th, and 7th-bit values from the pixel under consideration into the bit string M.	
         If SB equals 3	
          Sequentially store the 5th, 8th, 7th, and 6th-bit values from the pixel under consideration into the bit string M.	
         End of the conditional statement	
11      End of the conditional statement	
12     End of the nested loop for width	
13   End of the loop for height	
14   Compile each bit into an 8-column table, each row symbolizing a character’s bit within the concealed text.	
15   Leverage the powers of two to convert the collected bits into characters.	
16   Decide whether to employ the Vigenère or Caesar cipher on extracted bits.	
17   Display the secret text.	
18  End	

Receiver end

The receiver uses Algorithm 2, which retrieves the cipher text message from the stego image.

Experimental description and results

For the execution of this method, MATLAB 2018 is employed in the course of this investigation. The tests were performed on a system with an Intel Core i5 CPU running at 1.70 GHz and 8 GB of RAM, and other parameters are as follows: The suggested method employed 15 to encrypt the message with the Caesar cipher. The user can use any different number instead of 15 to encrypt data. In this study,

The proposed approach utilized the value 11 for the hash function, although alternative values can be substituted instead of 11.

Four standard grayscale photographs were used as the cover images to evaluate the proposed method. The names of these pictures are. Nature,

Jet,

Fishing Boat, and

Forest.

They are all 512 × 512 pixel resolution., as shown in Fig. 4.

Figure 4 Cover images.

(A) Nature, (B) jet, (C) fishing boat, image credit: Aysegul Alp, pexels, (D) forest, image credit: wallpaper by cafai0627 (from Wallpapers.com).

Qualitative analysis

Histogram analysis

Histogram analysis serves the purpose of understanding how the stego image relates to the cover image. Through histogram analysis, the imperceptibility of the stego image is efficiently demonstrated. Histograms from the cover and stego images (Figs. 5–8) reveal slight alterations, indicating higher imperceptibility in the stego images than the cover images.

Figure 5 (A) Cover image of nature, (B) histogram of nature, (C) nature stego image, (D) histogram of stego nature.

Figure 6 (A) Cover image of jet, (B) histogram of jet, (C) stego image of jet, (D) histogram of stego jet.

Figure 7 (A) Cover image of fishing boat, image credit: Aysegul Alp, pexels, (B) histogram of fishing boat, (C) stego image of fishing boat, (D) histogram of stego fishing boat.

Figure 8 (A) Cover image of forest, (B) histogram of forest, (C) stego image of forest, (D) histogram of stego forest.

Figure source credit: wallpaper by cafai0627 (from Wallpapers.com).

Quantitative analysis

Compared to cutting-edge techniques

The outcomes of the suggested technique are compared to those of methods presently in use using the assessment criteria Payload, MSE, Bits per Pixel (bpp), and PSNR. The MSE and PSNR are calculated as given in Eqs. (1) and (2). (1) MSE=1n∑i=1n⁡(yi−y^i)2

where MSE is a mean square error, n is the number of data points, yi is the observed value, and the predicted value is y^i. (2) PSNR=10×log10((2552)/(MSE))

Tables 1 through four compare several images of Nature, including jets, fishing boats, and forests. Compared to many other current strategies, the suggested methodology makes locating the bit utilized to store the message difficult. The proposed method is more secure than existing methods with low MSE since it can easily anticipate the bit pattern (Ch et al., 2015; Nasreen et al., 2022).

Table 1 Gray stego image of nature.

Algorithms	PSNR	MSE	Payload	Bits per pixels (bpp)	
Dhawan & Gupta (2021)	51.89	0.42	21,845	0.08	
Charan et al. (2015)	57.19	0.12	65,536	0.25	
Wade et al. (2019)	34.35	23.88	87,381	0.33	
Muhammad et al. (2015)	44.62	2.24	221,987	0.8	
Shehzad & Daǧ (2019)	52.17	0.39	262,144	1	
Muhammad et al. (2017)	26.98	130.34	262,144	1	
Kuo, Kuo & Wuu (2015)	46.57	1.43	314,572	1.2	
Liu, Chang & Chien (2017)	44.37	2.4	410,636	1.56	
Hong (2018)	34.92	21.1	524,288	2	
Setiadi & Jumanto (2018)	37.54	11.45	589,824	2.25	
Shehzad & Dag (2017)	37.85	10.66	778,567	2.97	
Tseng & Leng (2014)	33.82	26.98	789,270	3.01	
Islam et al. (2016)	38.85	8.4	838,860	3.1	
Proposed	38.14	9.96	262,144	4	
35.08	20.15	524,288	4	
33.28	30.53	786,432	4	
32.06	40.43	1,048,576	4	

Table 1 compares the newly suggested method and established LSB steganography techniques using a stego image for nature image. In this experiment, the authors used four different payloads, i.e., 262,144, 524,288, 786,432, and 1,048,576. The findings reveal that the proposed techniques can effectively conceal a substantial payload while maintaining a minimal increase in MSE, all within a secure framework. The proposed method is faster than other techniques because it uses no maintenance technique, which is time-consuming but does not improve the image quality much.

Table 2 compares the newly introduced technique and established LSB steganography methods using a stego image featuring a jet. Within this experiment, the researchers employed four distinct payloads: 262,144, 524,288, 786,432, and 1,048,576. The outcome shows that the suggested techniques can hide a high payload with a minor increase in MSE in a secure way. The proposed method is faster than other techniques because it uses no maintenance technique, which is time-consuming but does not improve the image quality much.

Table 2 Gray stego image of jet.

Algorithms	PSNR	MSE	Payload	Bits per pixel (bpp)	
Dhawan & Gupta (2021)	51.92	0.41	21,845	0.08	
Wade et al. (2019)	35.9	16.71	87,381	0.33	
Muhammad et al. (2015)	46.93	1.31	238,872	0.9	
Shehzad & Daǧ (2019)	52.17	0.39	262,144	1	
Muhammad et al. (2017)	31.97	41.31	262,144	1	
Kuo, Kuo & Wuu (2015)	47.82	1.07	288,358	1.1	
Setiadi & Jumanto (2018)	37.68	11.09	589,824	2.25	
Shehzad & Dag (2017)	38.15	9.95	778,567	2.97	
Khalind & Aziz (2013)	42.51	3.64	814,497	3.1	
Lu & Lu (2017)	37.79	10.81	824,756	3.15	
Swain (2016)	33.6	28.38	825,753	3.15	
Hussain et al. (2018)	33.84	26.85	1,024,983	3.19	
Khodaei, Sadeghi Bigham & Faez (2016)	33.84	26.85	939,592	3.58	
Proposed	40.88	5.30	262,144	4	
36.8	13.47	524,288	4	
34.60	22.51	786,432	4	
32.84	33.76	1,048,576	4	

Table 3 contrasts the proposed technique with established LSB steganography techniques for the stego image of a fishing boat. In this experiment, the authors used four different payloads, i.e., 262,144, 524,288, 786,432, and 1,048,576. The findings reveal that the proposed techniques can effectively conceal a substantial payload while maintaining a minimal increase in MSE, all within a secure framework. Compared to other approaches, the proposed method demonstrates swifter performance owing to its omission of time-consuming maintenance techniques. While these techniques do not significantly enhance image quality, the proposed method prioritizes efficiency.

Table 3 Gray stego image of fishing boat.

Algorithms	PSNR	MSE	Payload	Bits per pixels (bpp)	
Wade et al. (2019)	35.66	17.66	87,381	0.33	
Muhammad et al. (2015)	44.84	2.13	222,589	0.84	
Shehzad & Daǧ (2019)	52.12	0.39	262,144	1	
Muhammad et al. (2017)	31.97	41.31	262,144	1	
Kuo, Kuo & Wuu (2015)	48.51	0.91	288,358	1.1	
Setiadi & Jumanto (2018)	37.48	11.61	589,824	2.25	
Shehzad & Dag (2017)	38.38	10.3	778,567	2.97	
Lu & Lu (2017)	37.62	11.24	810,735	3.09	
Swain (2016)	32.47	36.81	812,646	3.1	
Khalind & Aziz (2013)	36.66	14.03	846,516	3.22	
Khodaei, Sadeghi Bigham & Faez (2016)	33.52	28.91	858,580	3.27	
Tseng & Leng (2014)	42.35	3.78	833,482	3.28	
Proposed	37.84	10.68	262,144	4	
34.89	21.07	524,288	4	
33.15	31.46	786,432	4	
31.92	41.72	1,048,576	4	

Table 4 comprehensively compares the newly introduced technique and well-recognized LSB steganography methods, focusing on a stego image set within a forest image. During the experiment, the authors investigated four distinct payloads: 262,144, 524,288, 786,432, and 1,048,576. The findings underscore the proposed approach adeptly conceals substantial payloads, with only a marginal escalation in Mean Squared Error (MSE) within a secure context. Noteworthy is the superior efficiency of the proposed method compared to its counterparts, eschewing time-consuming maintenance techniques that yield marginal image quality improvement.

Conclusion

The central goal of this study was to facilitate discrete communication while maintaining the confidentiality of information from other group members. Among available online resources, digital images emerge as optimal transmitters. The study focuses on encoding messages within images to achieve this objective. Various steganographic methods hide information within images, each with its intricacies. Encryption choices adapt to application requirements, from complete invisibility to concealing substantial messages. This study explains image steganography’s applications and methods. This work converted plain text into ciphertext and encoded it as an image using a 4 LSB technique built on a hash function. The LSB of the image’s pixel is used to place the secret text. Human eyes cannot distinguish between the original and final images since only the LSBs are changed. The outcomes of the proposed method demonstrate substantial stego image data capacity, commendable peak signal-to-noise ratio (PSNR), and low mean square error (MSE) compared to existing methods in the field. This approach offers a combination of robust security measures and ample payload capacity.

Supplemental Information

Supplemental Information 1 Code.

Click here for additional data file.

Additional Information and Declarations

Competing Interests

Author Contributions

Data Availability

Muhammad Asif is an Academic Editor for Peerj Computer Science.

Yazeed Yasin Ghadi conceived and designed the experiments, performed the computation work, prepared figures and/or tables, authored or reviewed drafts of the article, and approved the final draft.

Tamara AlShloul conceived and designed the experiments, performed the experiments, analyzed the data, performed the computation work, prepared figures and/or tables, authored or reviewed drafts of the article, and approved the final draft.

Zahid Iqbal Nezami conceived and designed the experiments, analyzed the data, performed the computation work, prepared figures and/or tables, authored or reviewed drafts of the article, and approved the final draft.

Hamid Ali conceived and designed the experiments, performed the experiments, analyzed the data, performed the computation work, prepared figures and/or tables, authored or reviewed drafts of the article, and approved the final draft.

Muhammad Asif conceived and designed the experiments, performed the experiments, analyzed the data, performed the computation work, prepared figures and/or tables, authored or reviewed drafts of the article, and approved the final draft.

Mohamed Jaward Bah conceived and designed the experiments, prepared figures and/or tables, authored or reviewed drafts of the article, and approved the final draft.

The following information was supplied regarding data availability:

The code is available in the Supplemental File.

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
