# Peer review of "Enhanced payload volume in the least significant bits image steganography using hash function"

_PeerJ Computer Science, doi:10.7717/peerj-cs.1606_

## Round 0.1 · original submission · Major Revisions

It is clear that there are several areas that need improvement in the paper. The author should focus on extending the Introduction section to provide more context, clearly state the motivation for the study, and address the identified gaps in the field. Additionally, a detailed explanation of the hash function's usage and aims is required, and the Experimental Results section should be expanded for more in-depth analysis. It is essential to avoid repetitive paragraphs when explaining the tables and include a comprehensive discussion on why the proposed method outperforms the compared methods. By addressing these points, the paper's overall quality can be significantly enhanced.

**Language Note:** The review process has identified that the English language must be improved. PeerJ can provide language editing services - please contact us at copyediting@peerj.com for pricing (be sure to provide your manuscript number and title). Alternatively, you should make your own arrangements to improve the language quality and provide details in your response letter. – PeerJ Staff

Reviewer 1 ·

Basic reporting

No comment (please see Additional Comments)

Experimental design

No comment (please see Additional Comments)

Validity of the findings

No comment (please see Additional Comments)

Additional comments

The manuscript misses many things.
1) The section 1 (Introduction) should be extended.
2) I have not seen any motivation to conduct this study.
3) What is the conclusion to be drawn from Section 2 (Related Work)? There should be some drawbacks/gaps, and this study should aim to overcome them.
3) People in this field may know the general functionality of using hash functions. However, its usage and aims differ from study to study. The functionality of the hashing used in this study and why and how it is used are not clearly explained. The relevant explanation should be extended in such a way.
4) Section 4, which should present experimental results, is very short and descriptive.
5) The paragraphs, word by word, used to explain Tables 1, 2, 3 and 4 are exactly the same. In a journal of this level, it is not appropriate to use repetitive paragraphs.
6) The other one of the most important missing in this study is discussion. A deep-enough discussion should be added, including an explanation of why the proposed method works better than the others you compared.

Reviewer 2 ·

Basic reporting

In this paper, the authors used the four Least Significant Bits for steganography to hide the message in a secure way using a hash function. Before steganography, the message is encrypted using one of the encryption techniques; Caesar cipher or Vigenère cipher. Since just the Least significant bits (LSBs) are altered, human eyes cannot foresee the difference between the original and stego images.

The overall structure and presentation of the paper is good, but there are few observations and comments for this paper:

1. In abstract the authors said that Before steganography, the message is encrypted using one of the encryption techniques; Caesar cipher and Vigenère cipher, but in Algorithm 1 they write Protect the message by utilizing the encryption method Vigenère cipher or Caesar Cipher. The authors need to consistent in whole paper.
2. The first paragraph of the introduction section is not clear. It is suggested to re-write the whole paragraph.
3. At end of the introduction, the authors describe their contribution, " The primary goal of the suggested data embedding approach is to enhance the payload and enhance the protection of confidential information in a specified cover picture with the least amount of modification, such that the cover image's quality is undetectable to the human eye." Writing their contribution in bullets is recommended because the sentence is very long and hard to understand.
4. IN introduction section, the authors describe different types of steganographic techniques. It is recommended that the authors write a sentence before describing these types; what are you telling?
5. In section II, the sentence is hard to understand: " Steganography (SSIM), (NCC), image fidelity (IF), and resilience are all crucial aspects to consider. The value of Stego images, payload, MSE, and correlation are all related difficulties. " So, it needs to rewrite this sentence.
6. The definition of Asymmetric-key cryptography needs to shift after the detail of VIGENÈRE CIPHER.
7. Improve the pixel quality of figures.
8. It is suggested that the authors must describe with detail both algorithms (sender side + receiver side).
9. There are many grammatical mistakes present in the paper. The paper must be proofread carefully.

Experimental design

1. Section IV-A suggested to re-write, as the heading is quantitative analysis, but the subheading is compared to cutting edge techniques.
2. On line 304, 312, 321, and 330, the authors write “The results show that the proposed techniques can hide a high payload with a minor increase in MSE in a secure way”. It is suggested that the authors must describe the actual values of MSE with detail.
3. On line 306, the authors said that it uses no maintenance technique, which is time-consuming but does not improve the image quality much. What mean of maintenance techniques.

Validity of the findings

1. The author needs to validate the result using steganalysis techniques.

Additional comments

Included in basic reporting

Reviewer 3 ·

Basic reporting

BASIC REPORTING:
- In this paper, the authors proposed a high payload secure technique for steganography. The authors encrypt the message before steganography. They used Caesar cipher and Vigenère cipher for encryption. They hide the data in the last four LSB in a circular way that is not detectable without a secure key.

The overall presentation of the paper is good, but some observations and comments are required to incorporate before acceptance.

1. The authors used many abbreviations, such as PSNR, MSE etc, in the paper. Using the full form of these abbreviations on first use is suggested.
2. On line 68, the authors said that in a specified cover picture with the least amount of modification, the cover image’s quality is undetectable to the human eye. They used two words for the cover image (cover picture and cover image). It is suggested to use only cover image throughout the paper.
3. In the paper, Apostrophes are improper, such as in lines 74 and 75, the sentence “It is essential to safeguard these systems and data since software packages’ complexity and new technologies’ development threaten data security, integrity, and availability”. It should be the package’s complexity and new technologies development.
4. The stego image and stego pictures are interchangeably used in the paper. It is suggested that the authors must use any one of these.
5. On line 256, the word transmitter and receiver sides should be replaced with sender and receiver.
6. An explanation of both algorithms is required.
7. Improve the quality of figures.
8. There are many grammatical mistakes present in the paper. The paper must be proofread carefully.

Experimental design

1. In section IV, It is suggested that the authors describe the parameters in bullets.
2. It is suggested that the authors must elaborate results in more detail.

Validity of the findings

- Already given in Basic reporting

---

## Round 0.2 · accepted · Accept

The authors have incorporated the required changes into the article according to the suggestions provided by the referees.

Reviewer 1 ·

Basic reporting

no comment

Experimental design

no comment

Validity of the findings

no comment

Reviewer 2 ·

Basic reporting

The comments have been incorporated accordingly.

Experimental design

no comment

Validity of the findings

no comment

Additional comments

The comments have been incorporated.

Reviewer 3 ·

Basic reporting

Corrections have been made. This is acceptable.

Experimental design

Corrections have been made. This is acceptable.

Validity of the findings

Corrections have been made. This is acceptable.

Additional comments

Corrections have been made. This is acceptable.